# Assessment of Benefits and Risk of Genetically Modified Plants and Products: Current Controversies and Perspective

**Bimal Kumar Ghimire** [1], **Chang Yeon Yu** [2], **Won-Ryeol Kim** [1], **Hee-Sung Moon** [1], **Joohyun Lee** [1], **Seung Hyun Kim** [1] and **Ill Min Chung** [1,*]

1  Department of Crop Science, College of Sanghuh Life Science, Konkuk University, Seoul 05029, Republic of Korea
2  Bioherb Research Institute, Kangwon National University, Chuncheon 24341, Republic of Korea
*  Correspondence: imcim@konkuk.ac.kr; Tel.: +82-10-547-08301

**Abstract:** Genetic transformation has emerged as an important tool for the genetic improvement of valuable plants by incorporating new genes with desirable traits. These strategies are useful especially in crops to increase yields, disease resistance, tolerance to environmental stress (cold, heat, drought, salinity, herbicides, and insects) and increase biomass and medicinal values of plants. The production of healthy plants with more desirable products and yields can contribute to sustainable development goals. The introduction of genetically modified food into the market has raised potential risks. A proper assessment of their impact on the environment and biosafety is an important step before their commercialization. In this paper, we summarize and discuss the risks and benefits of genetically modified plants and products, human health hazards by genetically transformed plants, environmental effects, Biosafety regulations of GMO foods and products, and improvement of medicinal values of plants by the genetic transformation process. The mechanisms of action of those products, their sources, and their applications to the healthcare challenges are presented. The present studies pointed out the existence of several controversies in the use of GMOs, mainly related to the human health, nutritions, environmental issues. Willingness to accept genetically modified (GM) products and the adoption of biosafety regulations varies from country to country. Knowledge about the gene engineering technology, debate between the government agencies, scientist, environmentalist and related NGOs on the GM products are the major factors for low adoptions of biosafety regulation. Therefore, the genetic transformation will help in the advancement of plant species in the future; however, more research and detailed studies are required.

**Keywords:** transgenic plants; genetic transformation; environmental effects; biosafety regulations; *Agrobacterium tumefaciens*; electroporation





## 1. Introduction

Present agriculture practices alone cannot solve food security, and eradicate malnutrition and hunger that exist globally [1]. Recent research reported that approximately 17.2% of the global population is lacked to the access of nutritious and sufficient food [2]. According to a survey, the present global rate of increase in crop yield is less than 1.7% and currently, the rate of increase in agricultural yield needs to be 2.4% to meet the world's demands for grains and to improve the nutritional quality [3]. FAO predicted the loss of arable land available for crop production from the current 0.242 ha to 0.18 ha by 2050 [4]. Conventional breeding creates a new population by intercrossing several lines with another parental line, in hopes of expressing one or more desired traits [5]. The conventional breeding process process have certain limitations such as sexual incompatibility, gene linkage, and the time involved in obtaining cultivars [6].

Genetically modified organisms (GMOs) are usually referred to as living organisms whose genetic makeup has been artificially manipulated by inserting new genes through

the process called the technology of recombinant DNA, or genetic engineering giving the plants new characteristics [7]. Genetically modified plants and products that emerged along with advanced biotechnology can contribute to the increase in agricultural production, and improved nutritional values that could relieve global food shortages [8]. Moreover, genetic transformation methods have generated the possibility of producing plants with desired traits in elite cultivars in considerably shorter time-frames with reduced gene linkage problems [9]. This technique not only provides desirable traits but also improves nutritional levels, transforming them into rich and healthy food items in both dicot and monocot plants [10]. Such technology is widely applicable in transferring genes from any organism to a great variety of plant species, from wild to cultivated species, thus preventing the natural barriers between species and reducing the time to obtain new varieties [11]. The use of these techniques in agriculture has also enabled the discovery of processes that involve the use of DNA techniques, which enable their propagation via cell and tissue culture in vitro, and in the production of transgenic plants to develop drug leads, and perform biosynthesis of functional compounds, enzymes, and hormones, blood substitutes, vaccines and antibodies [12], for the production of medicines, recombinants, and industrial products [13].

Agricultural products produced by genetic manipulation of crops such as soy, cotton, tomato, potato, canola, and corn, among others, have already been approved for marketing [14]. GMOs have great potential to solve the poverty of the global population, improve the nutritional value of crops, reduce environmental pollution, enhance medicinal values, and contribute to the sustainability of agriculture [15]. Despite the advantages of GMOs, there are widespread concerns about the biosafety of the products, causing the great concern regarding human health and environmental integrity, and political and regulatory issues [16]. The use of transgenics generates controversies related to possible risks to human health, such as; food allergies, antibiotic resistance, increase in toxic substances, more pesticides in food consumed, and also the lack of information on packaging labels [17]. Several arguments and intense debates have emerged from different forums analyzing potential benefits and possible risks associated with the cultivation and consumption of GMOs in terms of ethical, environmental, health, biodiversity, and religious issues [16]. It is, therefore, important to conduct the risk assessment of genetically modified plants and their products by making common regulatory methods before their release into nature and applications [18]. However, restrictions on transgenic plants exist and vary from country to country.

This study aims to analyze the possible harm and benefits of transgenic food according to the scientists working in the field, making people know what they are consuming. Moreover, the impact of transgenic plants on human health, the environment, and agriculture have been analyzed critically. This study will also take a look at the Gm biosafety and regulatory framework for GM foods in different countries. We will also take a look at the risks and controversies of GMOs

## 2. Literature Search Method

The literature presented in the study covers different social science field related to the GMOs. We conducted a mini survey of literature of GM plant and its products published in the journals (research articles and reviews paper) from 2000 to 2022. The search engines used in the collection of published papers (accessed on 1 October 2022) were Google scholar (http://www.google.co.kr), PubMed. The data base such as Scopus (https://ww.scopus.com, accessed on 1 October 2022) were used selection of identification of publications. All the downloaded papers were peer reviewed, English language, and related to GM products. Relevant published papers were searched in the Google scholar using the list of keywords (search terms). The search terms were organized in the following different groups: genetically modified organisms (GMOs), risk and advantages of GMOs, GMOs and biofortifications, phytoremediation, allergens, phytochemicals, GMOs and environmental and human health issues, and biosafety regulations, GMOs and controversies. The collected data were analyzed and illustrated to obtain the results based on the objectives of the present study.

## 3. Plant Genetic Transformation Methods

Hundreds of plants species have been successfully transformed by various genetic transformations and for numerous useful traits; however, these techniques have inherent problems and limitations, including the lack of an efficient plant regeneration system, low frequency of transformation, genotype specificity, low availability of genes of interest and biosafety, and time and labor-intensiveness [19]. The successful regeneration of transgenic plants requires two major factors: an efficient, rapid, reproducible regeneration system and an effective method for the integration of genes into the DNA of plant cells [20]. Foreign genes can be introduced into plant genomes by various methods, including biolistics, sonication, liposomes, viral vectors, transfer mediated by *Agrobacterium*, chemicals, silicon carbide fibers, floral dip method, microinjection, and microlaser treatment depending on the species to be transformed and types of explants used [21]. Among these, transformation mediated by *Agrobacterium*, electroporation, and biolistics, are the most commonly used methods for producing commercially released transgenic plants. Despite the limitations of transgenic plants, there has been a continuous increase in the production of such plants to improve the nutritional and medicinal value of crops.

### 3.1. Agrobacterium-Mediated Transformation of the Plant

Plasmids are the most commonly used vectors in the genetic transformation of plants (Figure 1). These vectors have an artificial T-DNA, into which different transgenes can be inserted and transferred to host plants [22]. *Agrobacterium tumefaciens* and *A. rhizogenes* have plasmid types Ti and Ri, respectively, both of which can be used for genetic transformation. The advantage of *Agrobacterium-mediated* transformation is that it possesses the natural ability to transfer and integrate transgenes into the host cell, transfer large segments of DNA with only minimal rearrangement, and possess the higher rates of genetic transformation efficiency, low copy number integration, and enable the transmission of integrated genes into progeny in a Mendelian manner [23]. This technique is applicable in both monocot and dicot plants, algae and fungi, human cells, and sea urchin embryos [24,25]. The limiting factor of this technique is the ability to regenerate the transformed tissues and the low transformant ratio; in addition, the size and complexity of the Ti and Tr plasmids also influence the rate of transformation [26].

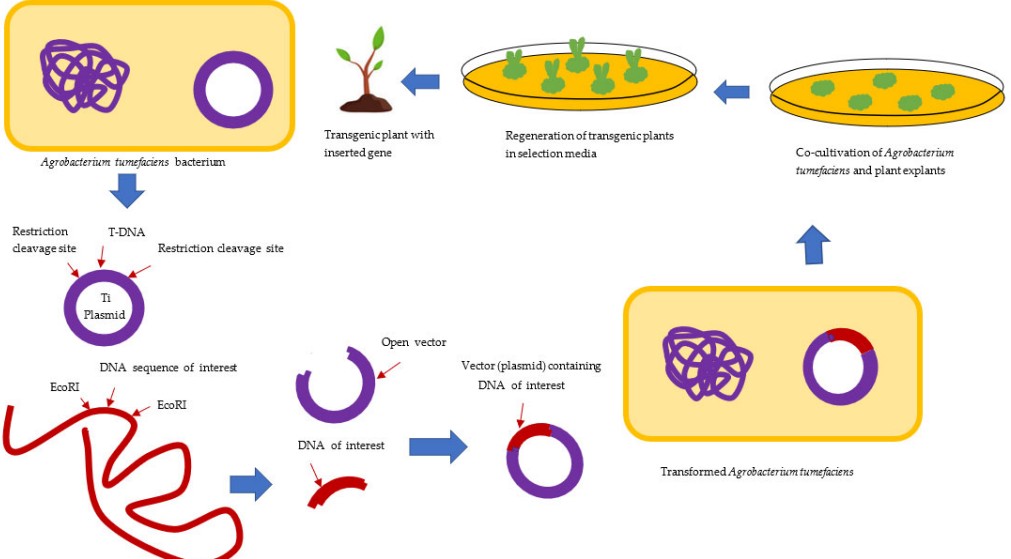

**Figure 1.** *Agrobacterium-mediated* genetic transformation of the plant. The schematic diagram shows the steps associated with the cloning of the gene of interest in the Ti-plasmid of *Agrobacterium tumefaciens* and its transfer to plant cells in culture to regenerate the transgenic plants with desirable traits.

### 3.2. Biolistics Method of Genetic Transformation of the Plant

This method is also called a gene gun, particle acceleration, or microparticle bombardment for the growth of transformants [27]. This method is useful for both dicot and monocot plants, consisting of bombarding cells or tissues with 0.5 mm gold or tungsten microparticles carrying exogenous DNA-coated projectiles using compressed helium incubated at 30 °C in a special chamber under vacuum conditions (Figure 2). The mechanism involves direct penetration of the cell wall and plasma membrane for direct DNA transfer [28]. Different systems are employed to accelerate the particles, including chemical explosion, higher pressure helium, electrical discharges, and vaporization of water drops [29]. This method readily transfers genes into intact plant tissues, including leaves, petals, and pollen endosperm, and has been successfully used to generate transgenic maize, soybeans, oats, rice, wheat, and barley [30]. The advantages of this method are that it consumes less time, allows the cells and tissues to undergo direct gene transformation, and can be applied to diverse groups of plant species with the higher stability of transformants [28].

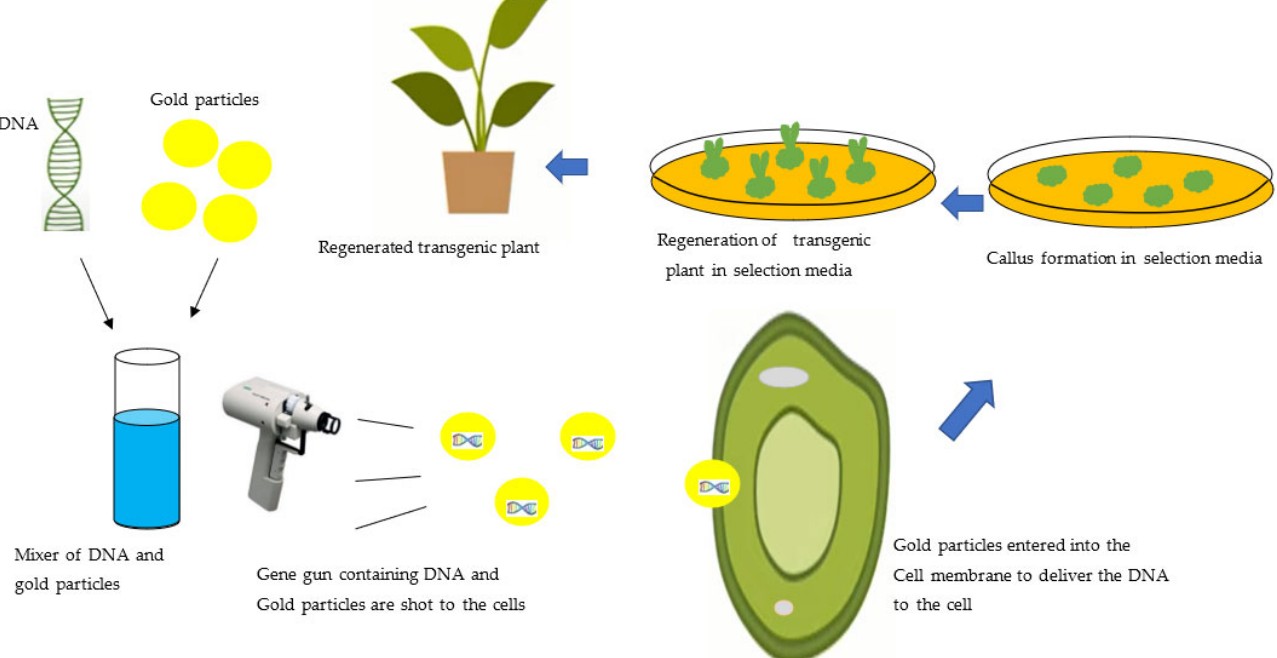

**Figure 2.** Biolistics method of genetic transformation of the plant. The schematic diagram shows the bombardment of gold particles containing the gene of interest onto the plant cells in culture to regenerate the transgenic plants with desirable traits.

### 3.3. Electroporation Method of Genetic Transformation of the Plant

The electroporation method was initially developed for the transformation of cereal genes and was later applied to other plant species. This method utilizes a high-voltage electric field to generate holes in the plasma membrane (Figure 3). The electrostatic forces formed in the process cause compression, which leads to the formation of holes in the membrane to integrate the transgene to be taken up by the cell [31]. The successful regeneration of transgenic plants using electroporation methods depends on various factors, including the diameter and source of host cells, electroporation medium (pH), electrical conductivity, membrane composition, size and shape of introduced DNA, and intensity and duration of electrical pulses used in the process [31]. The limitation of this method is the production of efficient protoplasm regeneration protocols and high cell mortality [32].

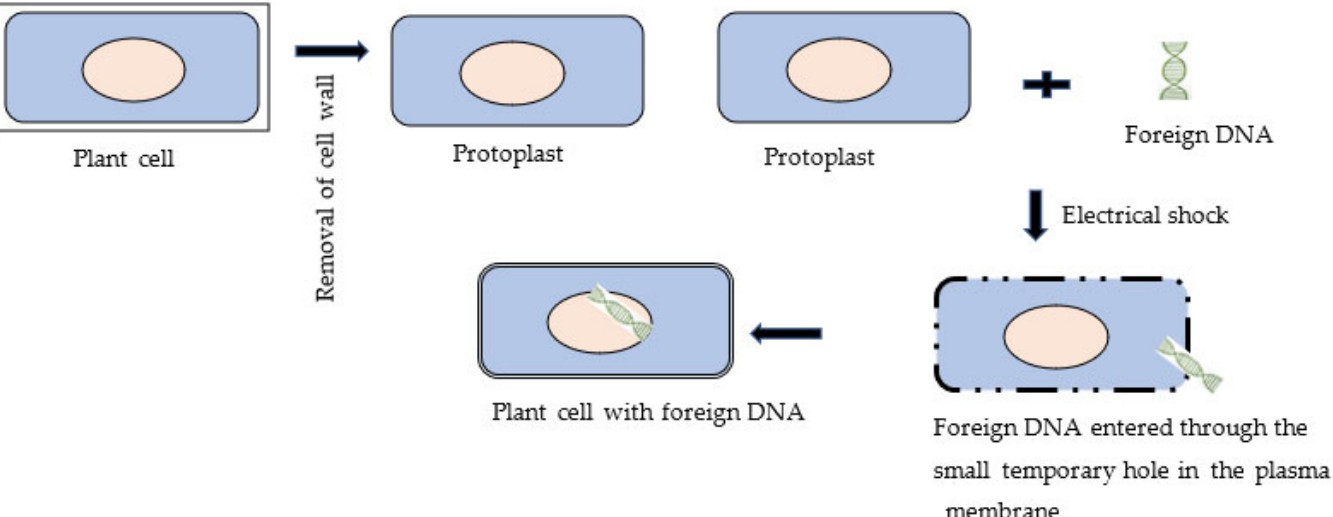

**Figure 3.** Electroporation method of genetic transformation of the plant. The schematic diagram shows the steps involved during the electroporation that lead to the insertion of the exogenous gene of interest into the plant cell.

## 4. Benefits of Genetically Modified Plants and Products

### 4.1. Biofortification

Micronutrient deficiencies are posing a serious threat to the health of one–half of the global population [33]. Nutritionally enhanced food crops using modern biotechnology, conventional selective breeding, and agronomic practices to enhance nutritional values are considered an effective and alternative approach for mitigating in economically poor countries [34]. The production of foods using biotechnology offers both benefits and threats. The production of transgenic plants is not only helpful in developing new varieties with increased nutrition but also increased resistance against biotic and abiotic factors, thereby enhancing the quality and yield of plants [35]. In addition, plant production enables the production of materials of industrial interest, such as biodegradable plastics, vaccines (transgenic bananas that produce vaccines against hepatitis B, transgenic potatoes that are resistant to viruses, rice with increased iron and vitamin levels, with increased resistance to extreme weather, and drought, [12,36,37].

GMO consumption maintains a healthy balance by fortifying nutritional quantity in foodstuffs that may not normally occur in them. For example, the production of "golden rice" with elevated vitamin A levels, the development of herbicide- and insecticide-resistant crops, thereby reducing crop losses, and other therapeutic substances of specific interest [38]. Moreover, research reports have indicated that proteins produced by GMOs are non-toxic, easily digestible, and cause no allergies [39]. Genetically modified fish grow larger, and pigs are grown with less body fat [40]. Other studies have reported increased beneficial nutritional profiles, such as increased levels of antioxidant compounds in GMOs that may provide health benefits to humans [41], and provide useful medicines, such as insulin for treating diabetes, from genetically engineered bacteria [42].

### 4.2. Transgenic Approaches for Improving Phytochemicals and Biological Activities in Plants

Several authors have reported an improvement in the production of antioxidants, such as phenolic compounds, from transgenic plants transformed with the bacteria *Agrobacterium tumefaciens* and *A. rhizogenes* (Table 1). Increased concentrations of phenolic compounds have also been reported to improve antimicrobial activities in *Cucumis melo* [43]. Furthermore, scientists have produced transgenic lines by overexpressing genes in *Lycopersicon esculentum* Mill. cv. Per with increased phenolic compound content in plants that are involved in phytoremediation [44]. Moreover, an increase in metabolites such as triterpene and steroidal saponins, and phenolics [45], was reported in hairy root cultures of

*Trigonella foenum-graecum* L., an elevated amount of phenolics acid, and flavonoids [46] was reported in *Spagneticola calendulacea* (L.) Pruski to increase food value. Increased resistance to *Botrytis cinerea* in transgenic *Morus notabilis* C.K. Schneid [47].

Genetic engineering has been successful in producing transgenic rice that contains 23 times higher concentrations of carotenoids than in previous transgenic golden rice [48]. Similarly, the genetic transformation of phytase in the transgenic soybean resulted in enhanced phytase activity by 2.5 fold compared to non-trangenic soybean [49]. Moreover, methyltransferase genes (*VTE3* and *VTE4*) from *Arabidopsis thaliana* transformed into the soybean genome resulted in an enhanced $\alpha$-tocopherol content by 95% more than in non-trangenic plants [50]. The transformation of lactoferrin in dehusked rice successfully enhanced the iron contents by 120% [51]. In another report, expression of soybean ferritin in rice resulted in an increase in the iron contents in Indica cv IRR68144 seeds, in wheat by 1.5–1.9 fold [52], lactoferrin genes enhanced the Fe content in Maize [53], potato, lettuce and tomato [54], Endogenous nicotianamine content was increased by 5–10 fold in transgenic rice over-expressed with HvNaSi [55]. Induces the proliferation of hairy roots, which increases the production of secondary metabolites. Many plant species have been transformed with *A. rhizogenes* for increased production of polyphenolic antioxidants such as phenolic acids and flavonoids (Table 1). Transformed plants of *Codonopsis lanceolata* and *Perilla frutescens* transformed with $\gamma$-*tmt* genes present higher concentrations of tocopherol and phenolic compounds, thereby enhancing the antioxidant properties of such plants [25,56]. Another approach to the recombinant production of foodstuffs is the genetic transformation of useful genes that enhance the production of beneficial compounds in plants and improve human health. Recently, researchers introduced genes into *Lycopersicon esculentum* Mill. cv Ailsa Craig, to increase the accumulation of antioxidants, such as phenolic compounds [57]. Similarly, increased amounts of phenolic compounds and resveratrol have been reported in transgenic *Rehmannia glutinosa* transformed by *A. tumefaciens* [58].



**Table 1.** Genetic transformation strategies and genes used for the biofortification in crops.

| Scientific Name | Agrobacterium Strains/Vector | Gene | Phytochemicals | Biological Activity | References |
|---|---|---|---|---|---|
| *Codonopsis lanceolata* | LBA4404/pYBI121, | *γ-tmt* | Phenolic compounds and tocopherol | Antioxidant and antimicrobial activity | Ghimire et al. [25] |
| *Perilla frutescens* | LBA4404/pYBI130 | *γ-tmt* | Phenolic compounds and tocopherol | Antioxidant and antimicrobial activity | Ghimire et al. [57] |
| *Lycopersicon esculentum* L. | pBI101 | *stilbene synthase (StSy)* | Resveratrol | Antioxidant activity | D'Introno et al. [59] |
| *Cucumis melo* | MAFF 03–01724 (pRi1724) | *rolC* gene | Aroma essential oils (Z)-3-hexenol, (E)-2-hexenal, 1-nonanol, and (Z)-6-nonenol | Antimicrobial activity | Matsuda et al. [43] |
| Wheat | pMDC32 | *Nicotianamine synthase 2 (OsNAS2)* | Higher concentration of grain iron and zinc | | Beasley et al. [60] |
| Cassava | LBA4404/p8023 | *FER1* and *IRT1* | Higher concentration of iron and zinc | | Narayanan et al. [61] |
| Rice | pMDC32 | *35S-OsGGP* | Increase concentrations of ascorbate | | Broad et al. [62] |
| Soybean | EHA105/pATPS1 | Overexpression of adenosine 5'-phosphosulfate sulfurylase 1 | Higher aamounts of sulfate, cysteine, and secondary metabolites in seeds | | Kim et al. [63] |
| *Gynostemma pentaphyllum* | ATCC 15834 | TL-DNA rolB | Triterpene saponins | Antitumor, immunopotentiating, antioxidant, antidiabetic | Chang et al. [64] |
| *Momordica charantia* | ATCC 15834 | *rolC* gene | Charantin | Antioxidant, antibacterial, antifungal | Thiruvengadam et al. [65] |
| *Momordica dioica* | KCTC 2703 | *rolC* gene | Phenolic compounds | Antioxidant, antibacterial. | Thiruvengadam et al. [66] |
| *Cucumis anguria* | KCTC 2703 | *rolC* gene | Phenolic compounds | antioxidant, antibacterial | Yoon et al. [67] |
| *Lycopersicon esculentum* Mill. | pBBC200/pBBC3 | *LC* and *C1*. | Flavonoids | Antioxidant activity | Le Gall et al. [68] |
| *Rehmannia glutinosa* | LBA4404/pMG-AhRS3 | Resveratrol Synthase Gene (*RS3*). | Phenolic compounds and Resveratrol | Antioxidant activity | Lim et al. [58] |
| *Ipomoea batatas* [L.] Lam. | pCAMBIA1300 | *IbCAD1* | lignin contents, monolignol levels, and syringyl (S)/guaiacyl (G) | Stress tolerance | Lee et al. [69] |
| *Miscanthus sinensis* | LBA4404/pMBP1 | antisense *COMT* gene. | Lignin content | Lignin biosynthesis | Yoo et al. [70] |
| *Cucumis melo* | MAFF 03-01724 | *rolC*gene | Volatile compounds | Antimicrobial activity | Matsuda et al. [43] |
| *Trigonella foenum-graecum* L. | ARqua1 and LBA9402, nary vectorp35S::eGFP, | Green fluorescent protein gene [eGFP S65T variant | triterpene and steroidal saponins, phenolics, and galactomana | Heterologous expression | Garagounis et al. [45] |
| *Sphagneticola calendulacea* (L.) Pruski | LBA1334, pCAM:2 × 35S:g | *rolA,rolB, rolC* and *gusA* | Phenolics acid and flavonoids | Anti-hepatotoxic activity | Kundua et al. [46] |
| *Morus notabilis* | GV3101/pLGNL | *MnMET1* | Flavonoid content | Inhibitory effect on *Botrytis cinerea* | Xin et al. [47] |
| *Arabidopsis thaliana* (L.) | pCAMBIA1301-AtMyB12 | *AtMYB12* | Phenolic compounds | Increase in the flavonoid contents | Wang et al. [71] |
| *Gynostemma pentaphyllum* | ATCC 15834 | TL-DNA *rolB* | Triterpene saponins (gypenosides) | Antitumor, cholesterol lowering, immunopotentiating, antioxidant, hypoglycemic, antidiabetic activity | Chang et al. [64] |
| *Aspergillus niger* | ANIp7-laeA | *LaeA* | flaviolin, orlandin and kotan | Biosynthetic model for flaviolin | Wang et al. [72] |
| *Nicotiana tabacum* | pCAMBIA1301- | *LlCCR* | Phenolic compounds, | Wood properties | Prashant et al. [73] |
| *Brassica rapa* ssp. rapa | KCTC 2703 | rolC and virD2 | Phenolic compounds | Antioxidant activity, antimicrobial activity | Chung et al. [74] |
| *Hypericum perforatum* L. | Ri plasmid | *rol*B | Phenolic compounds, hypericin, and pseudohypericin | Antioxidant activity | Tusevski et al. [75] |
| *Nicotiana tabacum* L. | pGANE7/pBAK61 | AK-6b | Phenolic compounds | Auxin and cytokinin | Galis et al. [76] |
| *Solanum tuberosum* | LB4404/pBinKan-TX | TyrDC2 | Phenolic compounds, tyrosol glucoside | Increased resistance against pathogens | Landtag et al. [77] |
| *Salvia miltiorrhiza* Bunge | GV3101/pHB-GFP | RAS and CYP98A1 | Phenolic compounds | Antibacterial; Antioxidant activity; | Fu et al. [78] |
| *Nicotiana tabacum* L. | LB4404 | ipt-Genes | Phenolic compounds | Peroxidase activity | Schnablová et al. [79] |
| *Artemisia carvifolia* Buch | GV3101 c/pPCV002 | *rol* Genes | Artemisinin | Increased production of artemisinin | Dilshad et al. [80] |
| *Cucumis anguria* L. | BA9402, A4, 15834, 13333, R1200, R1000 | *rol A* and *rol B* | Phenolic compounds | Antioxidant and antimicrobial activity | Sahayarayan et al. [81] |
| *Medicago sativa* | LBA4404 /pUC18-PAL | *COMT* and *CCoAOMT* | Phenolic compounds | Lignin biosynthesis | Guo et al. [82] |
| *Nannochloropsis* sp. | BA4404/pCAMBIA130404 | *gus–mgfp5* | Phenolic compounds | Transient GUS expression in | Cha et al. [83] |
| *Linum usitatissimum* | C58C1:pGV2260 | Chalcone synthase (CHS), chalcone isomerase (CHI), and dihydroflavonol reductase (DFR) | Phenolic compounds, monounsaturated fatty acids, and lignans content | Antioxidant properties | Lorenc-Kukuła et al. [84] |

### 4.3. Transgenic Approaches for Environmental Protection

The benefits of transgenics can be assessed from an environmental point of view (Table 2). *Bacillus subtilis* and *Bacillus thuringiensis* (Bt) strains can produce toxic proteins such as Cry or d-endotoxins [85], that are toxic to various kinds of pests, insects, and pathogens [86]. Bt toxins are also being used in generating trangenic crops effectively control crops pests such as CryIAc in rapeseed to control hairy bugs, diamondback moths, and cotton bollworms [87]. Cry2Aagene in transgenic pigeon beans to control pod borers [88], Cry3A gene in transgenic spruce to control bark beetles [89]. According to a recent report, a significant change in the amount of herbicides and pesticide application was observed in the USA with the adoption of herbicides tolerant GM plants [90], such as; transgenic soybean [91], summer corn and cotton. The reduction of herbicides and pesticides can reduce the environmental impacts on cultivated land. The reduction in the application of pesticides also minimizes the use of machinery for spraying them in the field, thus reducing fossil fuel consumption in the agriculture sector.

**Table 2.** Genetic transformation strategies and genes used for the improvement of biotic and abiotic stress resistance in crops.

| Scientific Name | Plant Parts | *A. tumefaciens* Strains/Vector | Gene | Biotic and Abiotic Resistance | References |
|---|---|---|---|---|---|
| *Medicago sativa* | Leaves and petiole | *Agrobacterium. tumefaciens* LBA4404/ AGL01/s GV101 | CRY3A (Bт Toxin) | Insect resistance | Tohidfar et al. [92] |
| *Oryza sativa* L. | Seed | Particle bombardment | *ITR1 gene* | Insect resistance | Alfanso-Rubi et al. [93] |
| *Glycine max* L. | Somatic embryo | Micro projectile bombardment | Viral coat protein | Soybean dwarf virus resistance | Tougou et al. [94] |
| *Jatropha curcas* L. | Leaves | *Agrobacterium tumefaciens* EHA 105 strain | Chitinase | Disease resistance | Franco et al. [95] |
| *Glycine max* L. | Leaves | *Agrobacterium tumefaciens* | CRY1A gene (TIC107) | Insect resistance | Macrae et al. [96] |
| *Gossypium hirsutum* var Coker | Seed | *Agrobacterium tumefaciens* (LBA 4404)/pBI121 | CRY1AB gene | Insect resistance | Tohidfar et al. [97] |
| Brinjal | Leaves | *Agrobacterium tumefaciens* LBA4404/pBI121 | CYSTATIN gene | Higher rate of inhibition of root-knot nematode in transgenic plant | Papolu et al. [98] |
| Kiwi fruits | Leaves | *Agrobacterium tumefaciens* LBA4404/pBin513 | *sbtCryIAc gene* | Resistance against *Oraesia excavate* | Zhang et al. [99] |
| *Camelina sativa* L. | Floral parts | *Agrobacterium rhizogenes* (pB172)/plasmid pKYLX71.1 | ACD*S*: ACC deaminase | Salinity tolerance | Heydarian et al. [100] |
| *Arabidopsis thaliana* L. | Seedlings | *Agrobacterium tumefaciens* GV3101/pBI121 expression vector | Transcription factor JCCBF2 | Freezing tolerance | Wang et al. [101] |
| *Camelina sativa* L. | Flower, stem, leaf, and root | *Agrobacterium tumefaciens*/pCB302-3 vectors | CsHMA3 | Heavy metals tolerance | Park et al. [102] |

### 4.4. Transgenic Approaches for Removing Allergens

Genetic transformation technology successfully incorporated genes in the plants responsible for encoding non-allergic proteins, and hypoallergenic crops, thus improving food protein equality [103]. A significant reduction in peanut allergies was reported by silencing the gene encoding Arah2 using RNAi technology [104]. Similar technology was used by Le et al. [105], to silence the allergens Lyce 1.01 and Lyce 102 in tomato profiling. Similarly, allergic proteins such as Mal d from apple [106], and GlymBd 30K from soybean [107] were silenced using RNAi technology. In other studies, the hypoallergenic approach was effective to reduce allergenic protein in Rye gram pollen [108]. All these studies indicate that engineered plants can also be expected to improve food quality by reducing allergens.

### 4.5. Transgenic Approaches for Phytoremediation

Phytoremediation is a sustainable solution for solving environmental contaminants caused by pollutants including heavy metals sediments, and inorganic and organic pollutants. Recently, the application of transgenic plants for the removal of heavy metals or organic pollutants has gained more interest [109] (Table 3). It is possible to transfer genes responsible for the hyperaccumulation of traits into target plants having remediation potential. The introduction of such genes has been reported in several plants including *A. thaliana*, [110]. Metallothioneins (MTs) confer heavy metal tolerance and accumulation in yeast. For example, the overexpression of MT genes increased the Cd tolerance in tobacco and raper seed plants [111]. Overexpression of *phytachelatin synthase* (TaPCSi) in *Nicotiana glauca* significantly increased the tolerance to heavy metals such as Cd and Pb [112]. In another study, overexpression of AtPCS, increased the phytochelatins and high resistance to arsenic [113]. Arsenate (As), mercury (Hg), and selenium (Se) are important pollutants, and transfer approaches have been employed to remove them from the soil [114]. Expression of the *mer B* gene in transgenic *Arabidopsis thaliana* resulted in more tolerance to methylmercury [115]. Similarly, overexpression of ATP sulfurylase and CGS resulted in an increased phytovolatilization in *Brassica* sp. [116]. Enzymes such as peroxidases, laccases, peroxygenases, nitroreductases, and phosphatases play important roles in the phytodegradation of organic pollutants [117]. These plant enzymes shown to act on organic pollutants including atrazine, chloroacetanilide, and TNT (2,4,6trinitrotoluene) [118]. An increased rate of degradation of TNT and chloroacetanilide has been reported previously in poplar plants [119]. Other best example of phytoremediation includes the overexpression of ECS and GS genes in *B. juncea* resulted in increased tolerance to atrazine [120].

**Table 3.** Genetic transformation strategies and genes used for increasing phytoremediation efficiency in crops.

| Plant | Gene | *A. tumefaciens* Strains/Vector | Product | Activity | References |
|---|---|---|---|---|---|
| *Arabidopsis thaliana* L. and Poplar | PtABCC1 | *A. tumefaciens* GV3101/pCX-SN | ABC transporter | Hg tolerance | Sun et al. [110] |
| *Arabidopsis thaliana* L. | TpNRAMP5 | pMD19-T, HBT95-GFP, pCAMBIA1305.1, | Numerous natural resistance-associated macrophage proteins | Increased accumulation of Cd, Co, and Mn | Peng et al. [86] |
| *Arabidopsis thaliana* L. | CsMTP9 | pENTR/D-TOPO vector into pMDC43 or pMDC83 | Metal transport protein 9 | Increased accumulation of Mn and Cd | Migocka et al. [121] |
| Tobacco | OsMTP1 | *E. coli, DH10B (GIBCO BRLp/UC18)* | Metal transport protein 1 | Cd hyperaccumulation | Das et al. [122] |
| *Salix matsudana* | ThMT3 | *A. tumefaciens* LBA4404/PROKII-ThMT3 | Metallothionein | Increased Cu tolerance and root growth | Yang et al. [123] |
| Tobacco | AtPCS1 | *A. tumefaciens* LBA4404/pBI121 and pCAMBIA | Phytochelatin synthase | Cd and As accumulation | Zanella et al. [124] |
| Petunia | RsMYB1 | *A. tumefaciens* C58C1/pB7WG2D | Transcription factor | Enhanced tolerant to Cd,, Cu, Zn | Ai et al. [125] |
| *Arabidopsis thaliana* L. | ZAT6 | *A. tumefaciens* GV3101/pXB93 | Zinc-finger transcription factor | Enhanced Cd tolerance | Chen et al. [126] |
| *Beta vulgaris* | St GCS-GS | *A. tumefaciens* EHA105/pGWB2 | StGCS-GS | Increased Cd, Zn, Cu tolerance | Liu et al. [127] |
| Rice | TaPCS1 | *A. tumefaciens* EHA105/pBI121 | Phytochelatin synthase, non-protein thiols | Cd hypersensitivity | Wang et al. [128] |
| *Arabidopsis thaliana* L. | AtABCC3 | *A. tumefaciens* GV3101/pER8 | Phytochelatin | Increased Cd tolerance | Brunetti et al. [129] |
| *Brassica napus* | BnNRAMP1b | ycf1 (Y04069), zrc1 (Y00829), smf1 (Y06272), BY4741/pYES2 | Transport functions | Enhanced uptake of Cd, Zn, Mn | Meng et al. [130] |
| Indian mustard | gshI, gshII and APS1 | pFF19 | γ-Glu-Cys synthetase, glutathione Synthetase, and ATP sulfurylase | Enhanced Se, | Banuelos et al. [131] |
| *Arabidopsis thaliana* L. | OASTd | *A. tumefaciens* CV50/pBI121 | Cysteine synthase | Tolerance to Cd | Dominguez-Solis et al. [132] |
| *Arabidopsis thaliana* | BnPCS | *A. tumefaciens* CV50/pBI121 | Phytochelatin | Tolerance to Cd | Bai et al. [133] |
| *Brassica napus* | CKX2 | *A. tumefaciens* GV3101 | Cytokinin content | Tolerance to Cd, Zn | Nehnevajova et al. [134] |

### 4.6. Transgenic Approaches for Vaccine Production

The expression of antigens using biotechnology in plants has opened up a new field for the production of plant-based vaccines. Advances in transgenic research have made use of plants to serve as a bioreactor for the production of certain vaccines for curing diseases [135]. Several plant-based vaccine antigens have been successfully expressed in plant tissues as a result of a stable expression or transient expression of genes [136] (Table 4). Plant-based vaccines are cost-effective, easy to carry, have less chance of contamination and degradation, require no medical professionals, high-tech machines, or preservation, and are less costlier than cell culture bioreactors [137]. By conceiving the idea of an edible vaccine, the antigens genes encoding Rabies Capsid proteins such as HBsAG, and HIVgag have been successfully expressed in transgenic tomatoes [138]. Exciting progress in achieving a high level of protein expression was achieved in transgenic carrots by Daniell et al. [139]. Later, Scotti and their research [140] team obtained chloroplast-based production of pharmaceuticals, vaccines, and antibodies. Transgenic *N. benthamiana* plants were successfully expressed with D antigen (PV3) to use a vaccine against polio diseases Marsian, et al. [141].

**Table 4.** Representative transgenic plant-based vaccines.

| Plants | Antigen/Virus | Diseases | Method of Administration | Reference |
|---|---|---|---|---|
| Transgenic potatoes | Hepatitis B surface antigen (HBsAg) | Hepatitis B | Oral | Richter et al. [142] |
| *N. tabacum* cv. Samsun | Virus glycoprotein and nucleoprotein fused with A1Mvcoat protein | Rabies | Parenteral | Yusibov et al. [143] |
| Potato, Maize kernels Potato | *E. coli* LT-B | Diarrhea | Oral | Tacket et al. [144] |
| Potato | Norwalk virus like particles (rNV) | Diarrhea, nausea | Oral | Mason et al. [145] |
| *N. benthamiana* | D antigen (PV3)/Poliovirus | polio | Intraperitoneal injections | Marsian, et al. [141] |
| *N. benthamiana* | H1, H5/Influenza virus | Influenza | NA | Makarkov et al. [146] |
| Peanut and tobacco | Glycoproteins hemaglutinin (H), Hemaglutinin neuraminidase (HS) | "cattle plague" and "Goat plague" | NA | Abha Khandelwal et al. [147] |
| *N. benthamiana* | VP2,VP3,VP5,VP7/African horse sickness virus (AHSV) | African horse | Intramuscular | Dennis et al. [148] |
| *N. benthamiana* | influenza HAC1 | H1N1 "swine" influenza | Intramuscular | Yusibov et al. [149] |
| *N. benthamiana* | Protective antigen (PA) | Anthrax | Subcutaneous | Watson et al. [150] |
| Maize | Spike protein | Swine transmissible gastroenteritis virus | Oral | Lamphear et al. [151] |
| Potato | CTB-gpl20 (HIV-1 gp 120V3 cholera toxin B subunit fusion gene) | Cholera | | Kim et al. [152] |
| Potato | HEV CP (HEV capsid proteins) | Hepatitis E | Oral | Maloney et al. [153] |

### 4.7. Transgenic Approach for Increased Biofuel Capacity in Plants

Lignocellulosic biomass from non-food crops has been considered a potential source of biofuel. Lignin, a major component of plant cell walls is considered a hindrance to cellulosic biofuel production. The application of biotechnology for biofuel production is gaining more interest, especially from the lignocellulosic biomass [154]. Recently, several studies have reported the successful cloning of genes responsible for increased biomass and sugar accumulation and higher production of biofuels in transgenic lines [155]. Several studies have reported the expression of genes in plants that are responsible for the degradation of the plant cell wall for more efficient biofuel production [156]. A low amount of lignin was reported by downregulating the lignin biosynthetic gene 4-hydroxycinnamoyl CoA ligase (4CL) [157]. In another study, the amount of lignin synthesized decreased to facilitate higher biofuel production in transgenic *Miscanthus sinensis* [70]. Overexpression of expansin genes which helps in loosening of cell walls [158] and successfully generated a transgenic plant with a suppressed debranching enzyme that produces soluble phytoglycogen. Vanden Wymelenberg et al. [156] reported the involvement of several genes in the breakdown of lignin from the *Phanerochaete chrysosporium* genome. Moreover, several other studies reported an alteration of lignin biosynthesis in the plant without affecting the vascular structure of plants [55]. They reported downregulating 4-hydroxy cinnamoyl CoA ligase

(4cl) responsible for the reduction of the lignin composition and an increase in the biomass of plants. Ralph et al. [159] reported a drastic decrease in the lignin content and structure by decreasing the expression of 4-coumarate 3-hydroxylase (C3H) in alfalfa. A similar result was also observed by Chabannes et al. [160] in transgenic tobacco by deducting the expression of cinnamoyl CoA reductase (ccR). Furthermore, the suitability of biofuel production in transgenic lines of tobacco has been investigated by downregulating O-methyl-transferase (OMT) enzyme by Blaschke et al. [161]. They observed an increase in biomass and reduction in the lignin contents in the transgenic lines of tobacco. Other emphasizes the improvement of the fatty acid composition of plants to enhance biofuel production. Moreover, as compared to the WT plants, the transgenic line showed an increase in biofuel production in soybean by expressing diacylglycerol acyltransferase 2A (DGAT2A) from *Umbelopsis* sps fungus [162]. Furthermore, an increased caprylic acid and capric acid was observed in transgenic rapeseed by over-expressing a laurate-specific LPAAT gene from coconut [163]. Another approach for increasing biofuel is to increase the biomass production of plants by genetic transformation approach. Manipulation of ADP glucose pyrophosphorylase resulted in an increased starch content and biofuel yield [164]. They observed an increase in photosynthesis and biomass by overexpressing two enzymes from Cyanobacteria in the tobacco plant. Jing et al. [165] reported an increase in the plant height and biomass by expressing the glutamine synthase gene (GSi).

*4.8. Increased Stress Resistance Capacity in Plants*

The excessive use of herbicides and pesticides is causing serious hazards on croplands, which makes cultivating land unsuitable for farming in the future. Recently, the introduction of GMOs has not required the use of these products. Some genetically modified crops are highly tolerant to one herbicide, instead of the multiple types of herbicides used in the field to prevent environmental damage. For example, genetically modified Roundup Ready corn is not only a glyphosate-tolerant GM corn but also is as safe and nutritious as conventional corn grain [166]. Bt rice KND1 expressing Cry1Ab protein show high levels of resistance to insects and possess no toxic effects on human health [167]. Similarly, insect-resistant crops include wheat, potatoes, rice, and sugarcane [168]. Researchers have increased the level of lignin content, monolignol levels, and syringyl (S)/guaiacyl (G) in transgenic *Ipomoea batatas* [L.] Lam., cv. Xushu 29 to enhance stress tolerance [69]. The introduction of Bt corn effectively controls the application of chemical pesticides, thereby controlling the environmental pollution caused by pesticides and reducing the cost of growing crops in the field [169]. Plants that can tolerate high salinity and long periods of drought have been reported [170], which can help people to grow crops in cold and less irrigated areas.

## 5. Disadvantages of Genetically Modified Plants and Products

The introduction of genetically modified food in the market has raised some serious questions regarding human health, environmental economics, and legal issues. For instance, it has been reported that the transfer of genes poses serious genetic hazards and is associated with possible food toxicity [41]. Once GMOs are produced and released into the environment, they can be difficult to control [171] and any harmful products produced by these organisms will remain metabolically active as long as they survive and multiply [171].

*5.1. Human Health Hazards*

Despite the advantages of GMOs, there is increasing concern about food safety and health risks. The transgene may cause undesirable developmental and physiological effects on mammals, including humans. There is a likelihood that the transformed gene may produce toxic protein or allergens or causes allergenic reaction in the human body. Moreover, other potential concerns are incomplete digestion of GMO foodstuffs in the gastrointestinal tract, which could result in the horizontal transfer of genes to the microflora

and somatic cells of the intestine [172]. Others have emphasized that the transfer of genes could cause infertility in animals, and result in allergic reactions [173].

### 5.2. Environmental Risks

The release of such products and their possible impacts on the environment regenerate high monitoring of environmental biosecurity to reduce or complete eradication of risk induced by them. Apart from direct effects on human health, GM plants have environmental effects on non-target organisms such as fish, worms, bees, and insects, biodiversity loss, and gene instability [174]. In other studies, Bt toxin produced by transgenic cotton killed many species of insect larvae, causing an imbalance in the ecosystem and food chain [175]. It has been argued that GM crops have a serious impact on farmers and their indigenous products because they compete with GMO products [176]. However, several previous studies reported the no-targeted impacts of novel genes transformed into the plant genome. For example, Bt maize showed potential hazards and toxic to monarch butterfly larvae that feed on milkweed leaves contaminated with pollens from Bt strains and caused delayed development, and increased maturity reported in *Ostrinia nubilalis* and *Spodoptera littorals* ingested with corn leaves expressing Bt CryIAS toxins [177].

### 5.3. Gene Flow

The most serious problem associated with gene flow is the loss of biodiversity and often cited as potential risk. Chances of accidental cross pollination between GM crops with its wild relatives are very high, making them super-weeds that resist diverse herbicides and become difficult to control. There are several examples where gene flow from crops to the relatives weeds such as in *Beta vulgaris* [178], in *Avena strigose* [179], in *Brassica napus* [180].

### 5.4. Increased Antibiotic Resistance

GM products enter the human body through food, vaccines, bacteria, or viruses. There is concern that the GM plants with bacterial resistance genes in their genome and might act as the source of drug resistance genes to the bacteria of clinical importance. Moreover, the possibility of developing antibiotic-resistant bacteria has been reported because of the frequent use of antibiotics in the genetic transformation process [181]. Most GM products contain marker genes and genes for certain useful traits. These marker genes can build resistance to particular antibiotics, and constant consumption of these foods could result in antibiotic resistance in the human body [182].

### 5.5. GMO Products Can Trigger Immune Reactions and Allergies

The introduction of new genes into plants can cause allergies by producing unexpected products (proteins and metabolites) in the plants [183]. For instance, the immune systems of rats respond more slowly to genetically modified potatoes than to normal plants [184]. In other studies, Bt bacteria can effectively control insects that attack crops. However, there is an equal chance of consuming Bt toxins and reacting to the mammals causing allergies [185]. Insects, birds, and other animals that feed on certain crops may not consume genetically modified crops due to allergic reactions or poisonous products. As a result, a great number of fauna can face starvation, affecting entire food chains and causing serious threats to ecosystems [186].

## 6. Biosafety Regulatory of GMO Foods and Products

Considering the importance of GMOs, several countries have managed to develop biosafety regulatory systems for the safety of GM foods and products. The regulations surrounding GMOs are complex and the rate of consumer acceptance is crucial, which results in reduced usage of GMOs. GMOs and their products have been facing severe controversies and hurdles from the public sector, NGOs, and environmental organizations [187]. Different governments have different approaches to tackling the products of GMOs, which vary widely, and are country-specific [188]. Within the European Union (EU),

Directive 2001/18/EL contains the biosafety regulation for the use of GMOs. It defines and control environmental release (case by case) evaluation of the environmental risk of GMOs [189]. Other directives such as 98/81/CE for the number of GM microorganisms, directive 1946/2003 for transboundary movement of GMOs, 1829/2003 for GM food and feed have been authorized [189]. GMO products have already been supplied to the EU market with appropriate labelling and identification methods under the title NOVWL-FOOD classification in May 1997 [190]. Currently, European Union-based legislation accepted the products of natural gene transfer methods, such as conjugation, auto-cloning, and gene transduction, and are considered non-genetically modified organisms [191]. However, EU has banned the application of clustered regularly interspaced short palindromic repeats genome (CRISPR-Cas9) editing technology, but the US has allowed the use of Cas9, which enables geneticists and medical researchers to edit parts of the genome [192]. Similarly, The Canadian Food Inspection Agency (CFIA) is responsible for regulating GM plants, a field trial of GM crops, their approval and commercial release in Canada. It also plays a major role in assessing impacts on biodiversity and environment, possible gene flow and impacts on non-targeted organisms [193]. In India, safety guidelines for GMOs such as research, field trails of GM foods and products assessment environmental risk assessment have been adopted from Rules 1989 [193]. Ministry of Environment Forest, Forest and Climate Change (MoEFCC) in association with the department of Biotechnology (DBT) recently adopted new guidelines for the environmental risk assessment of GE plants in India [194]. So far, Bt cotton (insect-resistant transgenic cotton) is the only GM plant to have been approved for commercial cultivation in India. Over 20 different GM plants with insect resistance, abiotic resistance, herbicidal resistance, enhance nutritional traits etc. have been under field trials [195].

The adoption of biosafety regulations is strongly impacted by the economical and political situation of countries. Despite their differences in approach and adoption of GMOs regulations framework, countries such as Brazil, Argentina, Chile, Mexico, Honduras, Costa Rica, and Uruguay were the first Latina America to approve GM crops [196,197]. Other Latin American nations such as Peru, Venezuela, and Ecuador implemented a complete ban on the application/test and import of GMOs [198,199]. To harmonize the regulations concerning GM products, Latin American countries such as Brazil, Argentina, Paraguay, Uruguay and Chile singed a declaration which legalizes the application of gene-edited products (case by case) amid strict regulation [200]. Countries such as Brazil and Argentina are major exporters of GM crops (cotton, soybean and Maize) and recently adopted legal provisions to allow the cultivation of GM crops [200], which not only play a bigger role in their economy but also play a key role to rapid adaption of biosafety law and regulations [200–203]. The Secretariat of Agriculture, livestock, fisheries and Food (SAGyO) is responsible for the regulation of GMOs, for conducting field tests, release and commercial application in Argentina [204]. While, national technological Biosafety committee (CTNBio), is responsible for scientific research on GMOs, field tests, risk assessment and assessing the safety of GMOs in Brazil [204]. Legal provisions of biosafety regulations are under discussion in the countries such as El Salvador, Mexico, Peru, Costa Rica, The Dominican Republic, and Ecuador. Other Latin American countries including Barbados, Dominica, Guyana, Haiti, The Bahamas, and Belize has no legal provision to deal with GMOs so far [205].

African nations can benefit from the adoption of the biosafety regulation to mitigate the food crisis, nutrition and economic livelihood [206–208]. The rapid adoption of GM crops regulations can address the existing food crises and ease hunger that exists in African countries. Some African countries welcomed GM technology and rapidly proceed for adopting GM crops to enhance agricultural production efficiency and increase the nutritional values of plants [209,210]. While, other African countries oppose GM technology stating its safety concerns, environmental and human health issues, intellectual property rights and ethical uncertainties [211–213]. However, several anti-GMO debates and controversies related to the safety of GMOs, and their impacts on human health and environmental issues

are major hindrances in adopting biosafety regulations among African nations [214,215]. Despite hindrances, the majority of African nations (47 countries) currently allow the cultivation of GMO crops [216]. South Africa is the first African nation to enact the regulatory framework to allow the cultivation, export and import of GM crops [216], and other African countries are interested in collaborating and harmonising the regulation concerning GM crops (African Biosafety network of expertise ABNE, 2019 [217]. Successful confined field trials have been conducted for maize, sorghum, cassava and Bt cotton with a wide range of traits in Kanya [217,218]. It has been reported that early acceptance of biosafety regulation has been hindered by inadequate GM technology knowledge in Kenya, and less awareness and knowledge of GM technology in the countries like Ghana and Nigeria, [219]. Moreover, a slow and delayed GM adoption rate in Tanzania have been reported [220]. The restrictive regulations, lack of information and awareness of the GM crops regulations have played an important role to obstruct the commercialization of GM crops in African nations [221,222]. In addition, opposition to biosafety bills, laws and regulations from NGOs, media, political parties social and economic factors and multinational companies have further helped to restrict the adoption of GM crops regulations in these countries [223–226].

Similarly, China adopts strict safety evaluation of GM plants and products and promulgated a whole set of biosafety laws, regulations and management systems considering its national situation and international norms and regulations. For the implementation of biosafety regulation, the Ministry of Agriculture (MOA) played a pioneering role in the implementation of regulations, and administrative Measures for the Safety Assessment of Agricultural GMOs [227], and developed the guidelines for safety inspection of field trials, research, processing, import and exports of GM crops [228]. Recently, MOA has promulgated a set of new regulations to shorten the process involved commercialization of GM crops [229] and introduced biosafety guidelines to regulate gene-edited crops [230]. Similarly, Korea has released a set of laws and regulations guidelines for GMOs and GMO products. To ensure biosafety, proper assessment of GMOs is carried out according to the guidelines of the Korea Food and Drug Administration (KFDA) [231]. It is clear from the above data that there exists a diverse range of regulations and frameworks supporting the research and commercialization of GM crops. For the efficient and successful functioning of these regulations, there is a need for a collective and synergetic approach, and closer interaction among the different government, non-government agencies, and private sectors which may play a diverse role in coordinating and harmonizing biosafety issues. Moreover, for adopting unified biosafety regulations, regional and international agencies should focus on the proper dissemination of information on biosafety regulations and public awareness about biosafety measures.

## 7. Controversies of GM Foods and Products

GMOs have become a controversial topic from the beginning. The supporters of GMOs including GM technologists, GM distributors, scientists and related regulatory agencies emphasize that GM products are non-toxic and nutritious [232,233], and potential to mitigate the global food crisis with no human health and environmental impacts. [234]. Moreover, several independent studies found no significant biological differences when GM crops and products were fed to the animals [235,236]. Some studies reported the presence of remnants of fragmented GM DNA in some parts of the gastrointestinal tracts, which were not detected in the blood and tissues [237,238]. Moreover, the in vitro experiment showed no horizontal transfer of GM DNA/genes to the microbes so far [237,238]. On the other hand, environmentalist opposes and rejected such results citing that the results were unacceptable due to methodological issues [190]. At the same time, opponents of GM believe that their exist differences between genetically engineered crops and traditional breeding plants. Moreover, Breckling et al. in their report pointed out a wide range of potential risks from GMOs including vertical gene transfer, horizontal gene transfer, hybridization, and resistance [239].

The risks potentials of transgene escape is high due to contamination in gene pool of crop landraces or wild relatives due to pollination of surrounding GM crops fields. Unwanted and unintended gene flow from the transgenic lines to wild relatives may produce genetically modified organism with unwanted traits that compete and displace the native species causing loss of genetic information [240]. Critics claims that the application of GMO can provoked the emergence of super weeds and pests that compel the use of more herbicides and pesticides to eliminate them from the field [240,241]. Moreover, various gene escape have been reported from oil rape to weedy relatives with glyphosate resistant trait [242], in creeping bent grass, in turf grass [243]. The transgene escape have been reported from Mexico in Maize landraces and cotton, that could change gene pool of maize landraces permanently [244–246]. A similar controversy has been reported in eggplant and its wild types [247].

The huge concern about the GMO is the corporate control of agriculture. Social activities from different parts of world believed that GM is private property, not national property [248]. The biotechnology companies has huge control over biotechnology process, genes and chemicals involve in the GM production process. As a result, handful of companies started protecting GM products, genes and chemical products through patents and licensing. For instance, Delta and Land Pine Company of Scott, Mississippi acquired patent on GM seed terminator that restricts unauthorized use of second generations' seeds, thus, consolidating its control over seed market for making huge profit. They claimed that seed terminator technique would solve the contamination of gene pool of relative wild plant species [249]. The sterile seed produced by the GM crops would not produce offspring. This will cause the non-availability crops seeds to the farmers and prevent farmers re-planting seeds. As a result, the farmer would be severely affected by patent rights, as they are required to sign contracts for replantation every year and timely seed supply and seed conservation [250]. Lured by high yields, farmer would quickly abundant traditional landraces causing huge loss in the biodiversity. Therefore, risks assessment of transgene escape and its possible consequence of recombination in plant genome, by monitoring the potential harmful effects on wild relatives is important steps in all the GM crops.

In a report from Chile, controversies regarding biosafety regulation have emerged from different sectors due to a lack of public access to regulatory information, and the location of GM fields or farm sites [251]. Anti GM campaign was initiated in 2011 and supported by coalition, organizations, farmers, "green" legislators, and anti-GM groups [251]. Similarly, GM policy debates in Ghana were initiated after enacting biosafety regulations related to GM (Biosafety Act 831, December 2011) [252]. Opponents of GMOs comprised of individuals, farmers, and civil society claimed that GM is discriminatory, with environmental and human health issues [252]. In Mexico, a social movement made up of indigenous, peasant, civil, cultural and scientific community organizations came together in an organized way in defence of Biosafety and Genetically Modified Organisms Law in the year 2005 [253]. Recently, the government has initiated a ban on GM maize and restricted the approval of new GM cotton seeds. Secretariat of Environment and Natural Resources (SEMARNAT) cited concerns about the possibility of genetically modified varieties being crossed with the native varieties of wild maize and cotton found in the country [253]. The rejection of GM cotton release permits has had a significant reduction on the cotton plantation (dropped by 30–35 per cent in 2020) and yield as growers can now only access poor yields of cotton with ineffective protection against pests on cotton varieties and impacted heavily on textile in Mexico. In some EU members such as Poland, the opposition to the distribution and cultivation of GM crops is as high as 60% [254]. EU ban on GM rice import from China was initiated after detecting GM rice in the tested sample. Illegal and large-scale planting and production of GM rice have been the practice before the certificate for GM rice issued by the Chinese government. As result, GM rice has been detected in China market without completing the proper experimental and biosafety test [255]. After detecting the GM contamination of rice, the EU blocked the import of GM rice (Bt Shanyo 63) to enter into its market and tightens its rules governing the imports of GM rice from China [256].

Other countries such as Russia, Israel, Norway and Netherland restricted the cultivation and commercialization of GM crops [256]. Other permissive countries such as South Korea, New Zealand, France and China have more restrictive regulations and permit the least number or no GM crops for commercial cultivation [257]. Similarly, a 2016 survey carried out in China showed about 47% of people held a negative view of GM crops [258].

## 8. Final Considerations and Future Prospects

Biotechnology is emerging with new opportunities for the production of food and energy, especially in countries where food production is still insufficient. Greater advantages of biotechnology will be established in the field of agriculture for the future demands of food security. Biotechnology can also help in generating plant species rich in cellulose for the production of biofuels, but also has many challenges. There are several advantages related to the genetic transformation of plant species and their application in improving medicinal value; plants resistant to abiotic and biotic stresses, plants with better nutritional value, and biomolecules important for industrial and therapeutic products. The introduction of biotechnology that introduces exogenous genes has made it possible for breeders to produce cultivars with improved genetic traits, which was not possible before. The growing global demand around these sectors is essential for the application of genetic transformation strategies for more plant species. The genetic transformation will help in the advancement of plant species in the future; however, more research and detailed studies are required. Despite the advantages of this technique, there is growing concern regarding the establishment of regulations for the efficient and safe use of GM plant products, and it is important to share knowledge concerning GM crops, including risks and benefits in terms of human health and the environment.

**Author Contributions:** B.K.G. contributed by writing the manuscript; C.Y.Y., W.-R.K., H.-S.M., J.L., S.H.K. and I.M.C. contributed by analyzing data and editing the revised manuscript. All authors have read and agreed to the published version of the manuscript.

**Funding:** This work was supported by the National Research Foundation of Korea (NRF) grant funded by the Korea government (MSIT) (NO. RS-2022-00165400).

**Institutional Review Board Statement:** Not applicable.

**Informed Consent Statement:** Not applicable.

**Data Availability Statement:** Not applicable.

**Acknowledgments:** This work was supported by funding from the KU Research Professor Program.

**Conflicts of Interest:** The authors declare no conflict of interest.

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
