# Peer review of "Assessment of Benefits and Risk of Genetically Modified Plants and Products: Current Controversies and Perspective"

_sustainability, doi:10.3390/su15021722_

Round 1

Reviewer 1 Report

Dear Authors,

 The submitted manuscript titled „Assessment of benefits and risk of genetically modified plants and products: Current controversies and perspective” presents very interesting results and therefore might interest the international audience. The topic of GMO is very interesting and publication summarizing the current state of knowledge is strongly needed.

However, I have found some imperfections, which (in my opinion) should be improved before an eventual publication. Please, find them below.

Major issue

In my opinion, the lack of description of method  of preparation literature survey in manuscript  is the strongest flawn. Such decription should justify why the publications were included into review. Which databases were used (if any)? Which time period was taken into account? The procedure of exclusion/inclusion of literature sources should be detaidly described.

I suggest to see the PRISMA statements in publications such as:

·         Moher D, Liberati A, Tetzlaff J, Altman DG; PRISMA Group. Preferred reporting items for systematic reviews and meta-analyses: the PRISMA statement. PLoS Med. 2009 Jul 21;6(7):e1000097.

·         Page et al. 2021. Updating guidance for reporting systematic reviews: development of the PRISMA 2020 statement. Journal of Clinical Epidemiology 134: 103–112.

Minor issues:

1.       Abstract section should reffer to particular parts of publication: background, methods, results and conclusions.

2.       In Introduction chapter the main goals should be listed.

Author Response

Reviewer 1

Reviewer’s comments

Dear Authors,

 The submitted manuscript titled „Assessment of benefits and risk of genetically modified plants and products: Current controversies and perspective” presents very interesting results and therefore might interest the international audience. The topic of GMO is very interesting and publication summarizing the current state of knowledge is strongly needed.

However, I have found some imperfections, which (in my opinion) should be improved before an eventual publication. Please, find them below.

Response:

Thank you for comments. We have revised the manuscript carefully and answered/modified/revised/re-write the manuscript as per the suggestions and guidelines of reviewers comments.

Reviewer’s comments

Major issue

In my opinion, the lack of description of method  of preparation literature survey in manuscript  is the strongest flawn. Such decription should justify why the publications were included into review. Which databases were used (if any)? Which time period was taken into account? The procedure of exclusion/inclusion of literature sources should be detaidly described.

 Response: We thank the reviewer for this suggestion.

“Methods of literature survey” included in the text.

Reviewer’s comments

I suggest to see the PRISMA statements in publications such as:

  • Moher D, Liberati A, Tetzlaff J, Altman DG; PRISMA Group. Preferred reporting items for systematic reviews and meta-analyses: the PRISMA statement. PLoS Med. 2009 Jul 21;6(7):e1000097.
  • Page et al. 2021. Updating guidance for reporting systematic reviews: development of the PRISMA 2020 statement. Journal of Clinical Epidemiology 134: 103–112.

Response: We thank the reviewer for this suggestion.

We have revised the manuscript and included the guidance given in the PRISMA statement.

Reviewer’s comments

Minor issues:

  1. Abstract section should reffer to particular parts of publication: background, methods, results and conclusions.

Response:

We thank the reviewer for this suggestion. Background, methods, results and conclusions indicated in the Abstract.

Reviewer’s comments

  1. In Introduction chapter the main goals should be listed.

Response: We thank the reviewer for this suggestion. Main goals included in the “Introduction”

Reviewer 2 Report

I’m sending you suggestions for manuscript corrections, please see below:

Page 2, Line 54: In the sentence: “Agricultural products produced by genetic manipulation of crops such as soy, cotton, tomato, potato, canola, and corn, among others, have already been approved for marketing”  please replace the word “produced” with the word “received”.

Page 2, Line 86: Please add a square bracket after the number 21.

Page 8, Line 213: Please add the full name of the plant “Spagneticola calendulacea (L.) Pruski”.

Page 8, Line 219: Please delete the redundant parentheses after Arabidopsis thaliana.

Page 8, Lines 232, 236, 238 and 241: I noticed an error in the reference numbers. In the line 232 you should refer to references 26 and 661, and in text line 236 to references 62, and in text line 238 reference 63. Please also renumber the mentioned references in Table 1 (for reference 63) and the bibliography.

Page 14: The paper number 116 – „Bhalla et al 2001“ is not cited in the text of the manuscript.

Page 14, Line 241: Please delete the redundant text “Phenolic compounds and tocopherol” from column about bilogical activity, second rows of tabele 1.

Page 14, Line 302: Please replace the word “mercury” with the chemical symbol “Hg” so that it is written uniformly with the other data in the Table 3.

Page 17:. The paper number 180 – “Blache et al 2004” is not cited in the text of the manuscript.

Take a note that there are some errors in References list. Please make a uniform style.

Author Response

Reviewer 2

Reviewer’s comments

Response:

Thank you for comments. We have revised the manuscript carefully and answered/modified/revised/re-write the manuscript as per the suggestions and guidelines of reviewer comments.

I’m sending you suggestions for manuscript corrections, please see below:

Page 2, Line 54: In the sentence: “Agricultural products produced by genetic manipulation of crops such as soy, cotton, tomato, potato, canola, and corn, among others, have already been approved for marketing”  please replace the word “produced” with the word “received”.

Response:

Thank you for comments. “produced” is replaced by “received”

Reviewer’s comments

Page 2, Line 86: Please add a square bracket after the number 21.

Response:

Thank you for comments. A square bracket included after the number 21.

Reviewer’s comments

Page 8, Line 213: Please add the full name of the plant “Spagneticola calendulacea (L.) Pruski”.

Response:

Thank you for comments. Full name of “Spagneticola calendulacea (L.) Pruski” included in the text.

Reviewer’s comments

Page 8, Line 219: Please delete the redundant parentheses after Arabidopsis thaliana.

Response:

Thank you for comments. The redundant parenthesis after the A. thaliana removed.

Reviewer’s comments

Page 8, Lines 232, 236, 238 and 241: I noticed an error in the reference numbers. In the line 232 you should refer to references 26 and 661, and in text line 236 to references 62, and in text line 238 reference 63. Please also renumber the mentioned references in Table 1 (for reference 63) and the bibliography.

Response:

Thank you for commentsReference number revised throughout the manuscript.

Reviewer’s comments

Page 14: The paper number 116 – “Bhalla et al 2001“ is not cited in the text of the manuscript.

Response:

Thank you for comments. References number 116 – “Bhalla et al 2001“ included in the manuscript.

Reviewer’s comments

Page 14, Line 241: Please delete the redundant text “Phenolic compounds and tocopherol” from column about bilogical activity, second rows of tabele 1.

Response:

Thank you for comments. Redundant text “Phenolic compounds and tocopherol” removed from the text.

Reviewer’s comments

Page 14, Line 302: Please replace the word “mercury” with the chemical symbol “Hg” so that it is written uniformly with the other data in the Table 3.

Response:

Thank you for comments. Mercury is replaced by Hg

Reviewer’s comments

Page 17:The paper number 180 – “Blache et al 2004” is not cited in the text of the manuscript.

Take a note that there are some errors in References list. Please make a uniform style.

Response:

Thank you for comments. Reference number 180 – “Blache et al 2004” included in the text.    

Reviewer 3 Report

 Assessment of benefits and risk of genetically modified plants 2 and products: Current controversies and perspective

 The paper by Ghimire et al., provides a review on Current controversies and perspective on GMOs. This is a highly relevant subject that is of great interest to various actors at the current time when the debate on GMOs rages on globally.

Here are my major concerns on the manuscript:

1.      Although the title of the paper is on controversies and perspectives, the paper does not highlight controversies on this subject but ends up discussing only the well-known disadvantages of GMOs. From the title I was expecting a balanced review of the views of various actors in the GMO debate and the controversies surrounding them. Instead of providing such an analysis, the paper provides literature on the GMO technology and its application, all of which are very well known and have been comprehensively reviewed over the years. For example, what are the issues that are leading some countries such as Mexico to abandon GMO and what are the controversies surrounding this issue? The GMO debate is both political and technical/scientific – what are the key issues on both fronts? The authors have to choose whether they want to review literature on GMO and its application or review the controversies and provide current perspectives. If they choose the latter, then GMO technology should be presented in some sort of an overview and not detailed review. There exists numerous comprehensive reviews on GMO technology and its application.

2.      The manuscripts reviews outdated literature, with almost 70% of the literature being over a decade old. Biotechnology is a very dynamic and fast changing field and concentrating on old literature denies the readers information on the current advances in the field. It is very important that the authors review recent literature for this paper to be interesting and of value to readers.

3.      Section 4 which appears to be the one that is most relevant to the title of the manuscript is poorly structured and highly repetitive. For example, the issue of nutritional composition of GMOs is repeated in both section 4.1, 4.2 and 4.4. Similarly, the effects of GMOs on human health are repeated in section 4.1 and 4.5. Gene flow is lumped together with issues of terminator gene instead of being discussed under environmental effects. Increased antibiotic resistance is lumped together with nutritional quality despite the two not being related. Nutritional quality should preferably have been discussed under effects of GMOs on human health. Section 4 is also full of sentences that are not clear and need to be reworded. These include line 425/426, 426/430 etc. Overall, this section needs to be rewritten afresh.

4.      Despite the title of the paper being “current perspectives…”, very little has been highlighted on this. The authors have provided a very superficial analysis of the available literature and very little on the authors perspectives.

5.      It is important to provide more information on overview biosafety frameworks globally not just in Europe and USA. This should include the biosafety frameworks during testing and commercialization of GMOs. In addition, it would be interesting to provide an overview of the status of GMO approval in different countries.

Other minor concerns include

·         Line 32/34: Instead of superior varieties, I would suggest that you talk of conventional breeding process - The conventional breeding process has certain limitations such as sexual incompatibility, gene linkage, and the 33 time involved in obtaining cultivars

·         Line 59: Change “most concern” to “great concern”

·         Line 306: Change “caring” to “curing” and revise the sentence as it is not clear.

·         Line 433: delete the word “food”

·         Line 435: Change the word “with” to “of”

·         Line 492: Change the word “participate” to “help”

Author Response

Reviewer 3

Reviewer’s comments

The paper by Ghimire et al., provides a review on Current controversies and perspective on GMOs. This is a highly relevant subject that is of great interest to various actors at the current time when the debate on GMOs rages on globally.

 Response:

Thank you for comments. We have revised the manuscript carefully and answered/modified/revised/re-write the manuscript as per the suggestions and guidelines of reviewer comments.

Reviewer’s comments

Here are my major concerns on the manuscript:

  1. Although the title of the paper is on controversies and perspectives, the paper does not highlight controversies on this subject but ends up discussing only the well-known disadvantages of GMOs. From the title I was expecting a balanced review of the views of various actors in the GMO debate and the controversies surrounding them. Instead of providing such an analysis, the paper provides literature on the GMO technology and its application, all of which are very well known and have been comprehensively reviewed over the years. For example, what are the issues that are leading some countries such as Mexico to abandon GMO and what are the controversies surrounding this issue? The GMO debate is both political and technical/scientific – what are the key issues on both fronts? The authors have to choose whether they want to review literature on GMO and its application or review the controversies and provide current perspectives. If they choose the latter, then GMO technology should be presented in some sort of an overview and not detailed review. There exists numerous comprehensive reviews on GMO technology and its application.

 Response:

Thank you for comments.

As per the author suggestion, we made a separate sub-heading for “controversies on GMO”.

Major issues on the ban on GMOs in the countries such as Mexico, EU has been included in the text. Debate both political and technical/scientific has been included.

GMO technology presented as overview.

Reviewer’s comments

  1. The manuscripts reviews outdated literature, with almost 70% of the literature being over a decade old. Biotechnology is a very dynamic and fast changing field and concentrating on old literature denies the readers information on the current advances in the field. It is very important that the authors review recent literature for this paper to be interesting and of value to readers.

 Response:

Thank you for comments. 35 outdated literature (published before the year 2000) has been replaced with recent literature.

Reviewer’s comments

  1. Section 4 which appears to be the one that is most relevant to the title of the manuscript is poorly structured and highly repetitive.

For example, the issue of nutritional composition of GMOs is repeated in both section 4.1, 4.2 and 4.4.

Similarly, the effects of GMOs on human health are repeated in section 4.1 and 4.5.

Gene flow is lumped together with issues of terminator gene instead of being discussed under environmental effects.

Increased antibiotic resistance is lumped together with nutritional quality despite the two not being related.

Nutritional quality should preferably have been discussed under effects of GMOs on human health.

Section 4 is also full of sentences that are not clear and need to be reworded. These include line 425/426, 426/430 etc. Overall, this section needs to be rewritten afresh.

 Response:

Thank you for comments.

Section 4.1, 4.2, and 4.4 revised and repeat sentences removed or replaced.

Similarly, repeat sentence exist in 4.1 and 4.5 deleted.

Terminator gene deleted from the section “gene flow and terminator gene”

In the revised version, “Nutritional quality” removed from the section “antibiotic resistance”

“Nutritional quality” discussed in the section “GMOs on human health”

Section 4 revised and lines 425/426, 426/430 corrected.

Reviewer’s comments

  1. Despite the title of the paper being “current perspectives…”, very little has been highlighted on this. The authors have provided a very superficial analysis of the available literature and very little on the authors perspectives.

 Response:

Thank you for comments.

Author perspectives included in the text.

Reviewer’s comments

  1. It is important to provide more information on overview biosafety frameworks globally not just in Europe and USA. This should include the biosafety frameworks during testing and commercialization of GMOs. In addition, it would be interesting to provide an overview of the status of GMO approval in different countries.

  Response:

Thank you for comments. In the revised version, biosafety framework from different countries from Asia, Africa, Latin Americas and EU included.

 Reviewer’s comments

Other minor concerns include

  • Line 32/34: Instead of superior varieties, I would suggest that you talk of conventional breeding process - The conventional breeding process has certain limitations such as sexual incompatibility, gene linkage, and the 33 time involved in obtaining cultivars
  • Line 59: Change “most concern” to “great concern”
  • Line 306: Change “caring” to “curing” and revise the sentence as it is not clear.
  • Line 433: delete the word “food”
  • Line 435: Change the word “with” to “of”
  • Line 492: Change the word “participate” to “help”

Response:

Thank you for comments.

Line 32/34: “superior varieties “ is replaced by “conventional breeding”

Line 59: “most concern” is replaced by “great concern”

Line 306: “caring” is replaced by “curing”

 Line 433: he word “food” deleted from the text

Line 435: “with”  replaced by “of”

  • Line 492: “participate” replaced by “help”

Reviewer 4 Report

This is a very interesting and important manuscript. The topic is something that causes controversies and it was very useful to read this review, on the advantages, disadvantages, benefits, and risks of GMOs. The list of references that the authors cited counts more than 200 titles, and the figures are representative and informative. Minor revisions are included in the text.

Author Response

Reviewer 4

Reviewer’s comments

This is a very interesting and important manuscript. The topic is something that causes controversies and it was very useful to read this review, on the advantages, disadvantages, benefits, and risks of GMOs. The list of references that the authors cited counts more than 200 titles, and the figures are representative and informative. Minor revisions are included in the text.

Response:

Thank you for comments. We have revised the manuscript carefully and revised the manuscript as per the suggestions and guidelines of reviewers and editor comments.

Line 122: underlined removed

Figure 1: Bottom part of the words corrected

Figure 2: Bottom part of the words corrected

Figure 3: Bottom part of the words corrected

Table 1: underlined removed

Line 432: Colon (:) removed

Reviewer 5 Report

The current manuscript is well written. This review is very sound and benefit for who want to read and have basic knowledge about plant transformation 

From my point of view, authors must focus on the plant transformation, GMO, only. 

Author Response

Reviewer 5

Reviewer’s comments

The current manuscript is well written. This review is very sound and benefit for who want to read and have basic knowledge about plant transformation 

From my point of view, authors must focus on the plant transformation, GMO, only. 

Response:      

Thank you for comments. We have revised the manuscript carefully and focused more on the GMO, biosafety framework and controversies of GMOs. 

Round 2

Reviewer 1 Report

Dear Authors,

In my opinion Your manuscript was sufficiently corrected, therefore I do not have any further remarks.

Author Response

Reviewers comments:

In my opinion Your manuscript was sufficiently corrected, therefore I do not have any further remarks.

Author response:

Thank you for providing an opportunity for revising the manuscript. We are also thankful for positive, supportive and sufficient guidance to bring this paper into proper shape and suitable for the publication.

Reviewer 3 Report

I wish to thank the authors for their efforts to revise the manuscript. However, I still have a few concerns

·         Based on the title of the manuscript, I would have expected greater focus on the controversies surrounding GMOs.  However, the manuscript is still heavily tilted towards GMO technologies. The section on controversies need to be expanded or the title changed to reflect the actual focus of the manuscript.

·         The manuscript has many grammatical and topographical errors that need to be corrected eg line 86- temrs, rish; line 23 – exixtence; line 82 and many more

·         Line 88: Is it really necessary to state the referencing software that was used???

·         Line 62: change received to produced

Author Response

Reviewers' comments: 

I wish to thank the authors for their efforts to revise the manuscript. However, I still have a few concerns

  • Based on the title of the manuscript, I would have expected greater focus on the controversies surrounding GMOs.  However, the manuscript is still heavily tilted towards GMO technologies. The section on controversies need to be expanded or the title changed to reflect the actual focus of the manuscript.
  • Author response: 

Thank you for providing an opportunity for revising the manuscript. We are also thankful for positive, supportive and sufficient guidance to bring this paper into proper shape and suitable for the publication.

We have expanded the section "controversies on GM plant and products".

Reviewers' comments: 

The manuscript has many grammatical and topographical errors that need to be corrected eg line 86- temrs, rish; line 23 – exixtence; line 82 and many more

  • Line 88: Is it really necessary to state the referencing software that was used???

  • Line 62: change received to produced
  • Author response: 
  • Thank you for the comments. Grammatical and topographical errors have been removed or corrected.
  • line 86- temrs, rish; changed to terms and risk, respectively
  • line 23 – exixtence; changed to existence
  • Line 88: Name of the referencing software removed from the text.